# Newly Discovered Alleles of the Tomato Antiflorigen Gene *SELF PRUNING* Provide a Range of Plant Compactness and Yield

**DOI:** 10.3390/ijms23137149

**Published:** 2022-06-28

**Authors:** Min-Sung Kang, Yong Jun Kim, Jung Heo, Sujeevan Rajendran, Xingang Wang, Jong Hyang Bae, Zachary Lippman, Soon Ju Park

**Affiliations:** 1Department of Biological Science and Institute of Basic Science, Wonkwang University, Iksan 54538, Korea; duke0357@naver.com (M.-S.K.); youngjun6381@naver.com (Y.J.K.); dandy3745@wku.ac.kr (J.H.); sujeevr@outlook.com (S.R.); 2Cold Spring Harbor Laboratory, Cold Spring Harbor, NY 11724, USA; xinwang@cshl.edu (X.W.); lippman@cshl.edu (Z.L.); 3Department of Horticulture Industry, Wonkwang University, Iksan 54538, Korea; bae@wku.ac.kr; 4Howard Hughes Medical Institute, Cold Spring Harbor Laboratory, Cold Spring Harbor, NY 11724, USA

**Keywords:** tomato, sympodial growth, semi-determinate, core collection, yield

## Abstract

In tomato cultivation, a rare natural mutation in the flowering repressor antiflorigen gene *SELF-PRUNING* (*sp-classic*) induces precocious shoot termination and is the foundation in determinate tomato breeding for open field production. Heterozygous *single flower truss* (*sft*) mutants in the florigen *SFT* gene in the background of *sp-classic* provide a heterosis-like effect by delaying shoot termination, suggesting the subtle suppression of determinacy by genetic modification of the florigen–antiflorigen balance could improve yield. Here, we isolated three new *sp* alleles from the tomato germplasm that show modified determinate growth compared to *sp-classic,* including one allele that mimics the effect of *sft* heterozygosity. Two deletion alleles eliminated functional transcripts and showed similar shoot termination, determinate growth, and yields as *sp-classic*. In contrast, amino acid substitution allele *sp-5732* showed semi-determinate growth with more leaves and sympodial shoots on all shoots. This translated to greater yield compared to the other stronger alleles by up to 42%. Transcriptome profiling of axillary (sympodial) shoot meristems (SYM) from *sp-classic* and wild type plants revealed six mis-regulated genes related to the floral transition, which were used as biomarkers to show that the maturation of SYMs in the weaker *sp-5732* genotype is delayed compared to *sp-classic,* consistent with delayed shoot termination and semi-determinate growth. Assessing *sp* allele frequencies from over 500 accessions indicated that one of the strong *sp* alleles (*sp-2798*) arose in early breeding cultivars but was not selected. The newly discovered *sp* alleles are potentially valuable resources to quantitatively manipulate shoot growth and yield in determinate breeding programs, with *sp-5732* providing an opportunity to develop semi-determinate field varieties with higher yields.

## 1. Introduction

Tomato is a major horticultural crop that continues to be intensively bred to improve cultivation-related traits, such as plant height, shoot determinacy, fruit size, and shape, through the selection of beneficial genetic variants [1,2]. Comparative genomics using large-scale genomic resources have identified many genes and alleles that were selected during tomato domestication and modern breeding [3,4,5]. Exploration of these genetic variations provides insight into the management of quantitative traits in tomato breeding [6]. For example, fruit size enlargement, a major domestication syndrome, originates in part from natural variations in *fasciated* (*fas*) resulting in the downregulation of the stem-cell repressing genes *SlCLAVATA3* (*SlCLV3*) and *locule number* (*lc*), resulting in an expanded expression domain of the stem-cell promoting gene *SlWUSCHEL* (*SlWUS*) [7,8]. The improvement of fruit mass under domestication combined the mutations in these two genes, which function in the conserved *CLV-WUS* negative feedback circuit [9]. Based on the molecular control of the *CLV-WUS* circuit, cis-regulatory variants of *SlCLV3* engineered by CRISPR-Cas9 genome editing recreated the effects of *fas* and *lc,* and further provided a continuum of fruit size, suggesting that this major domestication and breeding trait can be fine-tuned by expanding allelic diversity in these and other fruit size genes [10].

Genetic pathways that provided inflection points in cultivation-related traits have been well characterized in tomato, thus offering insight into the genetic and molecular changes that drove large-scale field cultivation [11,12]. For more than a century, *SELF PRUNING* (*sp*) mutants have precociously terminated tomato shoot growth naturally, and determinate tomato cultivars have been bred to have a shoot architecture suitable for mechanical fruit harvesting and fresh-market field production [13]. *SELF-PRUNING* (*SP*), a phosphatidylethanolamine binding protein (*PEBP*) family gene, was characterized as a positive controller of sympodial growth, which is defined by continuous cycling of vegetative-reproductive shoot units along primary and axillary shoots. This cycling is based on a balance of opposing flowering signals from *SP* and *SINGLE FLOWER TRUSS (SFT*) in axillary meristems [14]. Sympodial growth is completely inhibited in strong *sft* loss-of-function alleles, resulting in reversion of inflorescences to vegetative shoots with sparse production of flowers [15]. *SP* and *SFT* are homologs in the florigen gene family. Both of their encoded proteins bind to 14-3-3 adapter proteins in the cytoplasm of SAM cells and translocate into the nucleus, forming florigen activation complexes (FAC) through binding with the bZIP transcription factor SSP (ortholog of Arabidopsis FD) [16,17]. SFT induces the expression of floral transition genes such as homologs of *FRUITFUL1* (*SlFUL/TFUL)* and *SlFUL2/TFUL2* [18,19]. Conversely, SP was suggested to function in a similar system as FAC, but with the opposite function in tomato [17,20]. Therefore, the balance between *SFT* and *SP* expression is a crucial factor in determining the flowering time and growth pattern of tomato sympodial shoots. This balance model suggests that an advanced tomato shoot structure could be improved by finding a new balance between *SP* and *SFT* expression [12,15]. Indeed, hybrid studies of *sft* mutants showed that the partial suppression of shoot termination could increase yield by exploiting new balances of florigen signals in the *sp* background [21,22]. Moreover, the dosage sensitivity model of florigen was further supported using induced (EMS and genome-edited) mutations in *SP, SFT,* and *SSP* in various single-mutant higher-order mutant homozygous and heterozygous combinations [10,17]. For example, engineered promoter alleles of *SP* induced by the CRISPR/Cas9 system represented a continuum of quantitative variations for sympodial growth depending on the levels of *SP* expression, which is a new genetic resource to fine-tune and optimize plant determinacy and productivity in distinct breeding programs and agronomic conditions [10].

Since *sp-classic* has been known dominantly by marker-assisted selection breeding, only *sp-classic* was used for studying the interaction with modifiers from other genes/alleles that suppress or enhance sympodial growth. This means that we may have missed different *sp* alleles in the germplasm, which could be functionally stronger or weaker allele than *sp-classic*.

In this study, we hypothesized that germplasm resources provide an opportunity to find new *sp* alleles, potentially of different allelic strength, that would allow to find additional ways to tune architecture and yield/productivity. We then screened for and identified previously unknown genetic variants of *SP* from a collection of determinate genotypes in large tomato core collection (CC). By grouping the classic *sp* mutant and potential new *sp* alleles from 242 accessions phenotypes as determinate, we collected as determinate growth lines by genotyping classic *sp* mutants to isolate new *sp* alleles. The sympodial shoot growth and tomato fruit harvest of new *sp* alleles were carefully quantified compared to *sp-classic*. The molecular state of the new *sp* mutant was quantified using molecular markers to show that quantitative differences in determinacy are based on altered meristem maturation, which translates to differences in overall plant architecture and yield between four *sp* genotypes.

## 2. Results

### 2.1. Isolation of New sp Alleles from Core Collections

Recent reports have revealed that the suppression of *sp* can increase yields in determinate tomato cultivars [17,21]. We hypothesized that weak *sp* alleles could be isolated from *sp* allelic variations naturally collected or genetically modified (Figure 1), as a semi-determinate *sp* allele was selected among the *SP^CR-pro^* alleles [10]. 

To screen for possible new alleles of *SP,* we selected 242 core collections (CCs) pre-categorized as determinate or semi-determinate by Zamir’s lab. We genotyped them using the *sp*-*classic* CAPS marker. Genotyping this subset using PCR with the *sp*-*classic* marker revealed 52 CC lines that did not carry the *sp-classic* mutation (Appendix A). In contrast, 47 CC lines showed the same pattern as the *SP* wild-type genotype, 5 lines showed one band that was smaller than that of the *SP* genotype, suggesting a deletion event at the *SP* locus (Figure 2A). Furthermore, phenotyping of sympodial shoot growth validated that 14 out of 52 CCs were phenotypically determinate growth tomatoes (Appendix A). The remaining lines showed indeterminate growth, which was either due to the mis-categorization of determinacy or outcrossing with indeterminate genotypes in the open field. To test whether the 14 determinate CC lines were new *sp* alleles, PCR and Sanger sequencing of the *SP* promoter and coding region was performed. Five lines were shown to amplify a short PCR product 1 kb upstream and the first exon of *SP*, and also a 750 bp deletion in the *SP* promoter region from the middle of the first exon was identified (Figure 2B). Of the remaining lines, five lines amplified a short PCR product on the *SP* coding region including the 1st and 2nd exon and the 1st intron. This reflected that CCs have an identical 175 bp deletion in the 1st intron region (Figure 2C,D). Surprisingly, the last four CC did not show any difference between the PCR products in the *SP* promoter and *SP* coding region, but sequencing identified two synonymous substitutions on the second and third exons, converting lysine to glutamine (D99N) and alanine to aspartic acid (Q128K), respectively (Figure 2D,E). Notably, we also examined SNPs from 37 indeterminate CCs by Sanger sequencing and found that four CC lines carried a nonsynonymous mutation converting a valine to leucine (V99L; *sp-2446*) (Figure 2E, Appendix A).

In summary, along with complementation tests that confirmed all determinate lines were allelic for *sp-classic,* all determinate CCs were due to mutations in the *SP* locus: *sp-classic*, two deletional *sp* alleles and amino acid substitutional *sp* allele. To examine molecular differences in the three newly discovered *sp* alleles, *SP* expression and transcripts were examined using semi-quantitative RT-PCR from the SAM of 17 DAG seedlings that showed the promoter deletion allele. From these, *sp-2798* failed to produce *SP* transcripts, indicating a knockout allele (Figure 2F). The 1st intron deletion *sp* allele, *sp-2857*, accumulated two small *SP* transcripts in SYMs. Sanger sequencing revealed that all *sp-2857* transcripts were mis-spliced, suggesting that the intronic deletion caused a strong loss-of-function (Figure 2D,G). In contrast, while *sp-5732* showed relatively unchanged *SP* expression compared to WT plants, the two amino acid changes were in conserved positions among all SP orthologs (Figure 2E,F). In conclusion, based on the combined analysis of *SP* expression, transcript sequencing, and protein modification, the two deletion mutants *sp-2798* and *sp-2857* are predicted to knock out *SP*, whereas the amino acid changes in *sp-5732* are predicted to compromise SP protein function.

The discovery of three new *sp* alleles was surprising, especially given that all known determinate breeding lines and associated hybrid varieties are based on *sp-classic.* To determine when *sp* alleles arose and were largely used in breeding, we used resequencing data from 588 diverse tomato genomes to analyze the *sp* allele frequency. *sp-classic* was not detected in distantly related wild *Solanum* species or the progenitor species of domesticated tomatoes (*S. pimpinellifolium*), consistent with *sp-classic* first being documents nearly 100 years ago [13]. Indeed, *sp-classic* is only found in early domesticated genotypes (*S. lycopersicum var. cerasiforme*) and the ‘vintage’ accessions, which comprise cultivars that diverged approximately 75 years ago [23], reaching near-fixation in the processing cultivars. This suggested that *sp-classic* emerged during the early stages of modern breeding and were selected in modern processing cultivars (Figure 3, Appendix A). Interestingly, we found that new deletion *sp* allele was detected at low frequencies (0.1~0.2%) in accessions from wild relatives to modern tomato cultivars, indicating that *sp-2798* was not selected, and may remain in a state of drift as cryptic variants during tomato breeding (Figure 3, Appendix A), perhaps because its strong alleles result in too severe of a determinate growth habit. We also detected *sp-5732* with low frequencies (0.01~0.06%) in the first domesticated types and the ‘vintage’ accessions of the early domesticates and cultivars. Notably, this allele is completely absent in modern tomato cultivars, suggesting that marker-assisted selection may have eliminated *sp-5732* due to preference for a moderate effect on determinacy from *sp-classic* allele that breeders use for breeding fresh market and processing cultivars (Figure 3, Appendix A).

### 2.2. Comparisons of Shoot Structure and Yield Harvest among sp Mutants

To test the idea that the three *sp* alleles provide a range of determinacy, we performed comparative studies of sympodial shoot termination and yields. To directly compare shoot growth and termination among *sp* variants, the new *sp* alleles were introgressed into the processing cultivar M82 by backcrossing at least three times (BC2F2 or BC3F2) to establish near isogenic lines (NILs). We then quantified leaf numbers from the reference *sp-classic* and the new *sp* alleles on the primary shoot meristem (PSM) and successive SYMs on the main and axillary shoots. Shoot termination at the shoot apices of the three-month-old plants was also recorded (Figure 4A). Primary shoots accounted for seven to nine leaves in all *sp* alleles, indicating a similar flowering time in PSM among *sp* alleles (Figure 4B). *sp-2587* and *sp-2798* did not show much difference in the leaf number produced by the main shoot compared to the *sp-classic,* and no significant differences were observed for the leaf number produced in the axillary shoot compared to the *sp-classic* (Figure 4C,D). Although all shoots eventually terminated in the mature plants of all the *sp* genotypes, the leaf numbers of the main shoot and axillary shoot were higher in *sp-5732* compared to *sp-classic* (Figure 4C,D). These results indicate that overall flowering time and sympodial shoot cycling and termination is weaker in both primary shoot and axillary shoot systems compared to *sp-classic,* similar to the effects of *sft* and *ssp* heterozygosity (*sft-4537/+ ssp2129/+*, respectively) [17].

We next compared fruit yield among three *sp* variants; homozygous mutants from BC4F3 generations of *sp-2798* and *sp-5732* were compared with the control *sp-classic.* All three genotypes were grown under field conditions with controlled watering and nutrients. Shoot growth and termination of the genotypes grown under field conditions were identical to previous results, with more lateral organs and delayed flowering time and sympodial cycling and termination in *sp-5732* (Appendix A). Notably, the semi-determinacy of *sp-5732* translated to higher overall yield, with significantly increased plant mass, red fruits, and total fruit harvests, compared to other *sp* genotypes (Figure 5A–E). The Brix value, representing sugar content, was also increased in *sp-5732* (Figure 5F). The Brix yield of *sp-5732* increased by more than 50% with better fruit quality than that of *sp-classic* (Figure 5G). We validated these findings in a second yield trial, which showed a consistent yield increase of more 40% in *sp-4537* compared to the *sp-classic* in the open field (Appendix A).

### 2.3. Comparisons of Molecular States Using DEG Markers

Gene expression biomarkers are promising tools for better monitoring plant states for crop yield. To isolate the biomarkers for quantification of the molecular state of sympodial shoot meristems (SYM) between indeterminate and determinate growth, we compared the transitional meristems (TM) and SYM transcriptomes of *SP* and *sp-classic* profiled in our previous studies (Figure 6A,B) [17,24]. We previously defined TM as the stage of primary shoot meristem (PSM) switching to the reproductive phase and shows a broader and taller dome shape than vegetative stage PSM with a smaller last formed leaf (Figure 6A). SYM is a specialized axillary meristem in sympodial growth plant, which develops in the axil of the last leaf on the PSM and terminates after producing only three leaves under *SP* but producing zero to two leaves in *sp-classic* (Figure 6B). Notably, TMs and SYMs were collected and profiled at precisely matched morphogenic points in our previous studies (Figure 6A,B) [17,24]. Based on the normalized read counts of cDNA sequences (ITAG3.0), we identified 811 ‘TM DEGs’ between TMs of the genotypes, and 520 ‘SYM DEGs’ between SYMs of the genotypes; and a total of 984 ‘total DEGs’ between TM and SYM paired with genotypes using DESeq2 with cut-off criteria: log_2_fold-change ≥ 2, false discovery rate (FDR) < 0.05, and fragments per kilobase million (FPKM)/sample ≥ 3 (Appendix A) [24]. To explore the DEGs more deeply according to gene expression pattern, the DEGs were clustered into seven (I–VII) according to hierarchical expression in two tissues of the genotypes using hclust (Figure 6C; see Methods). Clusters I–IV (520 genes) were grouped as ‘Single DEGs’ containing genes differentially expressed in either TM or SYM. Cluster V–VII (464 genes) were grouped as ‘Co DEGs’ containing genes differentially expressed in both TM and SYM (Figure 6C, Appendix A). Therefore, these five groups of DEGs all indicate the patterns of DEG expression.

To biologically categorize each DEG into the five groups, we performed gene ontology (GO) enrichment analysis using the PANTHER classification system [25]. Single DEGs showed high enrichment for protein folding, developmental processes, single-multicellular organism processes, and multicellular organism processes GO terms in biological processes (Figure 6D). Specifically, the protein folding term was highly enriched in TM DEGs, but the remaining terms were enriched in SYM DEGs, reflecting tissue-specific functions. Notably, TM DEGs and Co DEGs were highly enriched in terms of biosynthetic processes, ribosome formation, and nuclear transport, reflecting overall differences in the development of *SP* and *sp-classic* (Figure 6D, Appendix A). Regarding molecular function, the terms for DNA binding, transcription, and DNA binding RNA polymerase were highly enriched in single DEGs and SYM DEGs, which is consistent with the enrichment of the nuclear term of the cellular component. This reflects a high difference in the regulation of transcription in SYMs between *SP* and *sp-classic* (Figure 6D, Appendix A). Altogether, SYM DEGs were functionally enriched in the developmental processes and transcription regulation, indicating SYM-specific functions.

To monitor the molecular state of *sp* alleles, we first isolated 55 SYM DEGs categorized under the development process and regulation of nucleic acid-templated transcription, which are major enriched GO terms, and then selected six biomarkers according to molecular functions related to the potential downstream genes of FAC, sympodial growth, and strong expression difference in *sp-classic* (Appendix A). The 55 DEGs were comprised of the potential targets of FAC, such as *SlFUL1*/*TFUL1*, *SlFUL2/TFUL2*, *MADS-BOX PROTEIN 20* (*MBP20*), the transcription factors (TF) controlling sympodial shoot and inflorescence structure at the post-transition stage such as *BLIND* [26], *LONG INFLORESCENCE (LIN)*, *JOINTLESS 2* (*J2*), and *ENHANCE OF JOINTLESS 2* (*EJ2*) [27], and the genes physically interacting with FAC targeting MADS TFs such as *RIPENING INHIBITOR* (*RIN*)*, J2,* and *EJ2* (Appendix A) [18]. Functionally unknown TFs, such as MYB, NT-FA, AP2, AT-hook motif, homeobox, and cold-shock-domain TFs were also characterized as highly downregulated or upregulated genes in the SYM of *sp-classic*. Therefore, two FAC downstream genes (*SlFUL2* and *MBD20*), two genes related to sympodial growth at the post-transition stage (*J2* and *EJ2*), and two functionally unknown and downregulated TFs (*Solyc12g009050* and *Solyc01g006930*) in *sp-classic* were selected as biomarkers for comparison of the molecular stages of *SP* alleles (Appendix A).

To quantify the molecular states in the SYM of *sp-5732* between indeterminate and determinate sympodial growth, we dissected the SYMs of *SP*, *sp-classic*, and *sp-5732* using a stereoscope (Figure 7A). Notably, meristem morphology of SYM was nearly indistinguishable among three *SP* alleles, with vegetative stages appearing as small dome structures producing the two leaf primordia (Figure 7A). The expression of biomarkers in the SYMs of *SP* and *sp-classic* provided a calibration stage between indeterminate and determinate sympodial shoot growth, thus enabling direct comparisons of the SYM states in *sp-5732* using real time quantitative RT-PCR (qRT-PCR) (Figure 7A–C). The expression patterns of all biomarkers were identical to the differences seen in the FPKM between *SP* and *sp-classic.* In the SYM of *sp-5732*, qRT-PCR results indicated that the expression of all biomarkers, including *SP*, was intermediate between the values of *SP* and *sp-classic*. *MBD20* and *SlFUL2/TFUL2* expression was upregulated to approximately half of the *sp-classic* value in the SYM of *sp-5732* (Figure 7D,E). *J2* and *EJ2* were weakly expressed and slightly upregulated in the SYM of *sp-5732* compared to that in *SP*, reflecting that the SYM of *sp-5732* is in the pre-transitional stage before sympodial growth and termination, like the wild type (Figure 7F,G). *Solyc12g009050* and *Solyc01g006930* were downregulated to nearly average values between *SP* and *sp-classic* in SYM of *sp-5732*, indicating that the biomarkers may play a role in the phase transition to the floral stage in SYM and confirming that the selected biomarkers successfully evaluated the maturation states of other types of SYM (Figure 7H).

## 3. Discussion

### 3.1. Allelic Variations in SP and Delay of Meristem Maturation and Sympodial Shoot Termination

An important goal of crop genetics and genomics is to identify all allelic variation in key productivity genes and establish quantitative genotype-to-phenotype relationships on growth and development to assess their value in breeding programs [28]. Natural variation in tomato *SP* and its homologs in many crops have been central in agronomic adaptions and yield enhancements in many crops, serving as inflection points that transformed indeterminate growth to determinate compact architectures. In tomato, this major change occurred within the last century, as is based on reduced SP activity mitigating the inhibition of florigen-based SFT signals in the sympodial shoot system. That all-determinate breeding in tomato appears to be based on the single allele of *sp-classic* suggests that breeders selected for a specific modified balance of florigen–antiflorigen (SFT-SP) signals to provide a level of determinacy that optimizes productivity in open field processing and fresh market production systems. Alternatively, *sp-classic* dominance in breeding may simply be serendipitous, and breeders have been fine-tuning determinacy with second site modifiers to adjust growth habit and yield over the last 100 years. In this regard, the prolonged SYM maturation—and thus extended sympodial cycling—of *sp-5732*, unlike other *sp* alleles, could be an as yet unrealized variant to improve yield by providing predictable breeding of semi-determinate growth.

Screening for *sp* alleles using determinate CC lines revealed three new *sp* alleles. Two deletional alleles were knock-out mutants showing no expression of *SP* in *sp-2798*, and expressing shortened and truncated transcripts due to mis-splicing of *SP* transcripts in *sp-2587.* Amino acids substituted in *sp*-*5732* and *sp-classic* were highly conserved in CETS proteins but were not key amino acids binding with 14-3-3 and were not distinct FLOWERING LOCUS T (FT)and TFL1-like homologs, indicating that structural variations of SP could indirectly and negatively affect the functions of antiflorigen (Figure 2E). Although we could not clearly explain the differences in functional defects among *sp* alleles, including between *sp-5732* and *sp-classic*, allelic variations in sympodial structure showed that *sp-5732* produced more leaves in each SYM and could produce more sympodial shoots in the plants than other *sp* alleles, reflecting that *sp-5732* delayed maturation in SYMs and termination of sympodial cycling is due to a weak allele effect relative to the other three alleles. Thus, *sp-5732* is similar to CRISPR-Cas9 engineered *SP* promoter alleles that provided a continuum of sympodial shoot termination, from modified indeterminate growth to semi-determinate and determinate growth [10]. Specifically, one *sp* promoter allele resulted in semi-determinate growth, also producing more leaves than the *sp*-*classic* and occasionally re-initiated sympodial shoots on primary and side shoots [10]. In support, comparisons of sympodial leaf production and biomarker expression among all natural *sp* alleles indicate that *sp-5732* has a prolonged vegetative phase during sympodial shoot maturation compared with that of *sp-classic* (Figure 4 and Figure 5).

### 3.2. Molecular Changes in Sympodial Shoot Termination

The molecular basis of local SFT/SP balance in sympodial shoot termination is the target competition system in which SP competes with SFT to bind with 14-3-3 to assemble FAC [20]. Our transcriptome analysis isolated SYM-specific upregulated genes containing *SlFUL1/TFUL*, *SlFUL2/TFUL2*, and *MBD20* in *sp-classic*, which are known as the potential direct targets of FAC, reflecting anti-FAC (SP:14-3-3:SSP) suppressed FAC target genes and phase transition (Figure 7D,E). Significant upregulation of *J2, EJ2, LIN,* and *RIN* in the *sp-classic* indicates that the SYM stage takes place during or after transition as *J2, EJ2,* and *LIN* are markers of sympodial inflorescence development, and *RIN* functions in fruit ripening after the transition to the floral stage (Appendix A). Conversely, two novel *NF-YAs* were significantly downregulated in *sp-classic*, indicating that NF-YAs play a role in the suppression of floral phase transition. These results suggest that the overexpression of *NF-Y* subunits such as *NF-YA, NF-YB*, and *NT-YC* alters flowering time and that NF-Y complexes regulate floral transition by highly redundant and complicated mechanisms in *Arabidopsis* [29]. In this respect, the state of SYM-producing second leaf primordium was molecularly evaluated as a state during or after floral transition in *sp-classic*, reflecting the precocious termination of the sympodial shoot, whereas wild-type SYM is still in the vegetative stage, indicating indeterminate reiterations of sympodial shoots.

### 3.3. Crop Performance through Modulation of the Antiflorigen/Florigen Signals

Late-flowering Micro-Tom variants rearranged with a continuum of flowering time indicated the scope for genetic resources with high biomass and fruit harvest under *sp* background [30]. Our yield trials using the newly discovered *sp* alleles and *sp-classic* indicated that moderate suppression of *sp* determinate growth enhanced the harvest of biomass and fruits. *sft* mutations as homozygotes genetically suppress sympodial shoot development in the main and axillary shoots of *sp* mutants, indicating the complete epistasis of *sft* over *sp* [15,31]. This epistasis is dosage dependent, as *sft/+* heterozygotes partially suppress *sp-classic* determinacy. Homozygous *ssp* mutants that disrupt FAC activity also suppress *sp* determination into indeterminate growth of the main and axillary shoots, as do heterozygotes in a similar dosage-dependent manner as *sft* alleles [17]. Across all these single and double mutant homozygous and heterozygous genotypic combinations, the flowering time of sympodial shoots was delayed, as reflected, for example, in the delayed expression of downstream MADS TFs expression, which are targeted by FACs [18]. Furthermore, double hybrids genetically combined with FAC components showed a progressive increase in overdominant fruit harvest due to an even greater quantitative suppression of sympodial shoot cycling and termination, suggesting that mutations in the florigen pathway could provide a broad toolkit to boost crop productivity [17]. As elaborated above, engineered *SP* promoter alleles mimic *sft/+* and *ssp/+* effects on determinacy, providing another route to achieving semi-determinate growth and potentially higher-yielding cultivars [10]. Here, we have added to this toolkit by identifying the new amino acid substitution *sp-5732* allele, which showed greater biomass and fruit harvest under field growth conditions.

Finally, our *sp* allele frequency assay indicated that the *sp-classic* was mainly used for breeding modern tomato cultivars. However, other alleles have cryptic variations that are hidden resources of natural variations. We reported that *sp-4537* improves fruit harvest under determinate growth. We strongly suggest breeders take advantage of *sp-4537* and other alleles to cultivate high-yielding modern processing tomatoes with optimizing plant determinacy.

## 4. Materials and Methods

### 4.1. Plant Materials and Genotyping

Tomato seeds of 242 CC inbred lines were obtained from Zamir’s lab at the Hebrew University of Jerusalem. All plants were grown using a water irrigation system at the farm of the Cold Spring Harbor Laboratory and Wonkwang University from April to July. *sp-classic* genotyping markers were used to screen for wild-type *SP* among the CC lines (Appendix A). *SP* CDS were amplified and sequenced to isolate the other *sp* alleles. To transfer new *sp* alleles into the tomato cultivar M82, the CC lines were backcrossed more than three times with cultivar M82. The new *sp* alleles were genotyped using their genotyping markers (Appendix A). The progeny of BC3F2 or BC2F2 were used for phenotyping, and the progeny of BC3F3 and BC3F4 were used for all yield trials in this study.

### 4.2. Tissue Collection, RNA Extraction, and RT-PCR

To extract total RNA from *sp-classic*, *sp-2798*, and *sp-5732,* shoot apices were collected between 13 and 17 days after germination (DAG) from plants grown in a greenhouse. As defined in a previous report [24], transitional meristems (TM) and sympodial shoot meristems (SYM) were imaged using a stereoscope. More than 30 meristems were dissected and collected from shoot apices fixed by acetone fixation for RNA stabilization, as previously reported [24]. Total RNA was extracted using the PicoPure RNA Extraction kit (Arcturus) and treated with the RNase-Free DNase Set (Qiagen, Valencia, CA, USA), according to the manufacturer’s instructions. One microgram of total RNA was used for cDNA synthesis using ReverTra Ace-α^®^ (TOYOBO, Osaka, Japan). RT-PCR was performed using i-Taq^TM^ DNA Polymerase (Intron) and a T100^TM^ Thermal Cycler system (Bio-Rad, Hercules, CA, USA). Real-time quantitative RT-PCR (qRT-PCR) was used to verify the expression of biomarkers in the *sp* alleles. Two biological replicates of TM and SYM were used for qRT-PCR, and the expression values were analyzed using the CFX96TM Real-time PCR System (Bio-Rad, Hercules, CA, USA). The threshold cycle (Ct) values were calculated and normalized against *Ubiquitin* using the StepOne™ software v2.3 (Applied Biosystems, Ltd., Waltham, MA, USA). Primer information is shown in Appendix A.

### 4.3. Yield Trials under Agricultural Field and Greenhouse

Yield trials were conducted in the Department of Horticulture Industry and in the tomato field at Wonkwang University in 2017 and 2019, as previously described [17]. Seedlings were grown in a greenhouse for 35–40 days and transplanted to the field and greenhouse at the beginning of April. Yield experiments were conducted under wide (1 plant per 0.75 m^2^) between plants using one irrigation and fertilizer regimes. Each genotype of the *sp* allele was represented by at least 16 biological replicates in the first trial and 11 replicates in the second trial. All the plants were transplanted in a completely randomized design. Damaged or diseased plants were excluded from the analyses.

### 4.4. Statistical Analyses of Yield Related Traits and Flowering Time

Harvesting was conducted in the middle of August in 2017 and 2019, when the majority of plants in a trial had over 80% red fruit. Phenotypic measurements of total fruit yield per plant, total number of inflorescences, and plant weight were taken after plants were manually removed from the root. Red and green fruits were collected as mature and immature fruits, and the total fruit yield was the sum of each plant’s red and green fruits. Ten fruits were randomly selected to estimate the average fruit weight and total soluble solids content (mainly sugars). The latter was referred to as the Brix value and was measured using a digital Brix refractometer (ATAGO). Brix yield was the total Brix content in the total yield per plant. Mean values for each measured yield parameter were analyzed using the “Fit Y by X” function and statistically compared using a Tukey-Kramer multiple comparison test, Dunnett’s ‘compare with control’ test, or t-test, whenever appropriate. At each time point, individual replicate plants were dissected for selected component traits of yield (plant weight, total fruit yield, Brix value, fruit weight, inflorescence number, and flowers per inflorescence). Flowering time was indirectly measured using the leaf number produced by the primary and sympodial shoots. Shoot termination was decided using the main shoot 30 day after transplantation. Shoot termination and flowering time were analyzed using data from a minimum of 12 biological replicates for each genotype.

### 4.5. RNA-seq Analysis

The RNA-seq data published in previous studies [17,24] were used for the analysis of differentially expressed genes between TM of *sp-classic* and wild-type plants, and between SYM *sp*-classic and wild-type plants. The paired-end reads of 2 replicates in *SP*_TM, *sp_*TM, *sp_*SYM, and *SP*_SYM were downloaded from the following address (*sp_*TM, *sp*_SYM, and *SP*_SYM, https://solgenomics.net/ftp/transcript_sequences/by_species/Solanum_lycopersicum/libraries/illumina/LippmanZ/; *SP*_TM, SRP090200 (SRA) (accessed on 27 June 2022). Low-quality reads were filtered using Trimmomatic_v0.36 software [32] for primer, adapter, and low-quality sequences. The following parameters were used: ILLUMINACLIP:path/Trimmomatic-0.36/adapters/TruSeq2-PE.fa:2:30:10, LEADING:3, TRAILING:3, SLIDINGWINDOW:4:15, MINLEN:36. Finally, the filtered reads were confirmed to be of high quality using FastQC v0.101.1 software (https://www.bioinformatics.babraham.ac.uk/projects/fastqc/, accessed on 27 June 2022). All filtered reads were aligned against annotated cDNAs from tomato ITAG3.0; http://solgenomics.net/organism/solanum_lycopersicum/genome, accessed on 27 June 2022) using the short-read mapping software Bowtie v2.26 [33], and the abundance of each transcript was estimated with FPKM values using RSEM v1.2.31 [34] with default parameters. The reads of the *sp_*SYM were split and aligned. The genes in our samples and the raw read counts of the genes served as the foundation for further analysis.

### 4.6. Differential Expression Gene and Gene Ontology Analysis

Statistical tests of differential gene expression based on raw read counts between each pair of samples involving two stages of M82 tomato meristems (TM and SYM) were conducted using R. Replicates were used in a modified exact test implemented by DESeq2 v1.26.0 [35] to test differential expression in two sample comparisons based on raw read counts. The *p*-values attained by the Wald test were corrected for multiple testing using the Benjamini and Hochberg’s procedure. As a result, differentially expressed genes achieved greater than log2 two-fold change with FDR < 0.05, and then genes showing less than average 3 FPKM/sample (summed over all stages) were removed. The filtered DEGs with FPKM values were hierarchically clustered using ‘hclust’ from the stats R package (R Core Team, 2019). To biologically categorize each DEGs of the five groups, GO enrichment analysis was performed using the PANTHER classification system (http://geneontology.org, accessed on 27 June 2022) with DEGs showing the expressions of log2 over four-fold change in the filtered DEGs, and enriched GO terms were selected with *p*-value < 0.01.

## Figures and Tables

**Figure 1 ijms-23-07149-f001:**
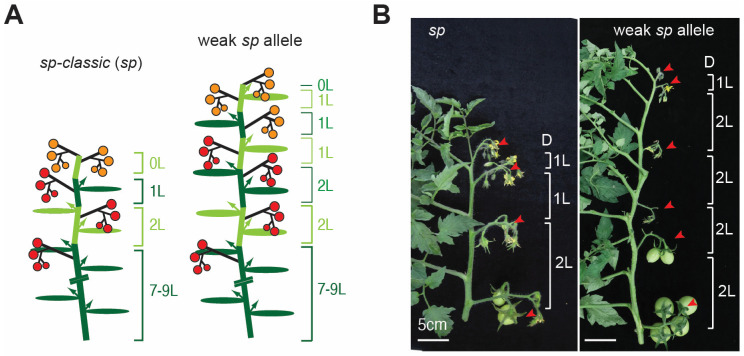
Tomato plant architecture of *sp-classic* and hypothetical weak *sp* allele. (**A**) Diagrams depicting tomato plant architecture of *sp-classic* and hypothetical weak *sp* allele. The dark green bars and ovals represent the primary shoot and associated leaves in the main shoot. Alternating white green and dark green bars and ovals indicate successive sympodial shoots. Arrows indicate axillary shoots and black lines indicate inflorescences. Red- and orange-colored circles represent maturing fruits. (**B**) Representative main shoots from *sp-classic* and hypothetical weak *sp* allele. Three-month-old plants are shown. Arrowheads indicate inflorescences. L, leaf. Scale bars, 5 cm.

**Figure 2 ijms-23-07149-f002:**
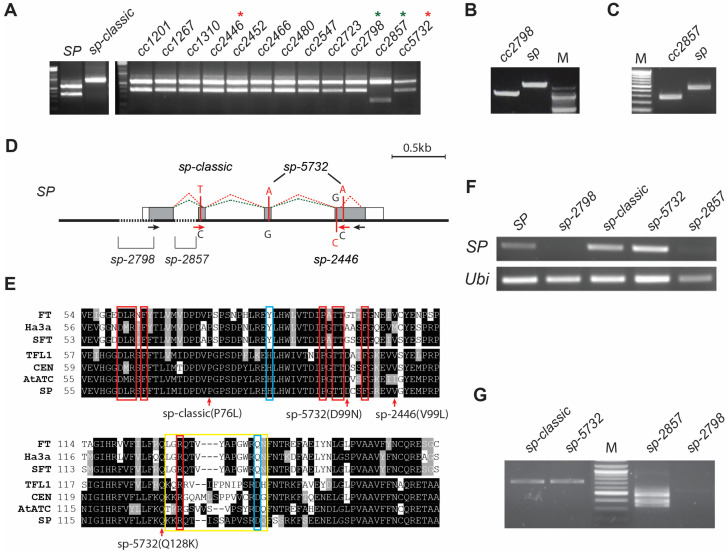
Identification of new *sp* alleles from core collections: (**A**) Genotyping by *sp-classic* CAPS marker. PCR products were digested by ScrFI cutting. *SP* and *sp-classic* are controls. Red asterisks, amino acid substitution *sp* mutants; green asterisks, new deletion *sp* mutant in *SP* locus. (**B**,**C**) Gel shift of PCR product amplifying 1kb *SP* promoter including the first exon (**B**) and amplifying the first exon and intron in *SP* locus (**C**). (**D**) Diagrams showing the *SP* gene structure and the locations of *sp* mutations. Dashed black lines indicate deletion regions of *sp* mutant. Dashed green and gray lines indicate mis-splicing of SP transcripts. red fonts, nucleotide substitution; white boxes, UTR; gray boxes, exon. (**E**) Partial alignment of SP homologs showing the external loop domain (yellow line box), residues binding to a 14-3-3 protein (red line boxes), and cue residues for florigen/antiflorigen function (blue line boxes). red arrows, sites of mutation substituted amino acid. (**F**,**G**) Semi-quantitative RT-PCR analysis of *SP* expression (**F**) and PCR amplification of full-length *SP* transcript including ORF (**G**) in *sp* mutant alleles and wild type (*SP*) at the shoot apex of SYM stage. *Ubiquitin* (*Ubi*) transcripts were used as PCR control. M, size marker.

**Figure 3 ijms-23-07149-f003:**
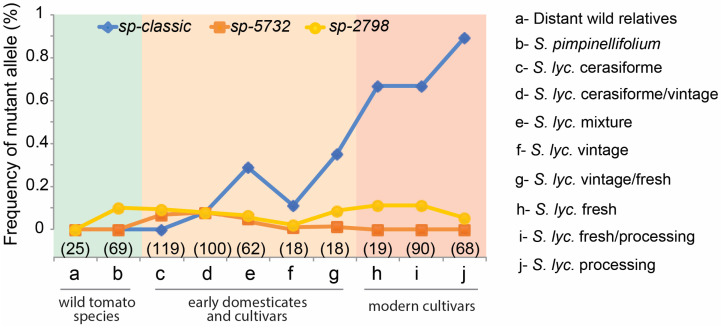
Allele frequencies of *sp* alleles in accessions classified as wild *Solanum* species (distant relatives and *S. pimpinellifolium*, the wild progenitor of domesticated tomato), early domesticates and cultivars (*S. lycopersicum* var. cerasiforme and *S. lycopersicum* vintage), and modern cultivars (fresh-market and processing). Number of accessions is indicated in parentheses.

**Figure 4 ijms-23-07149-f004:**
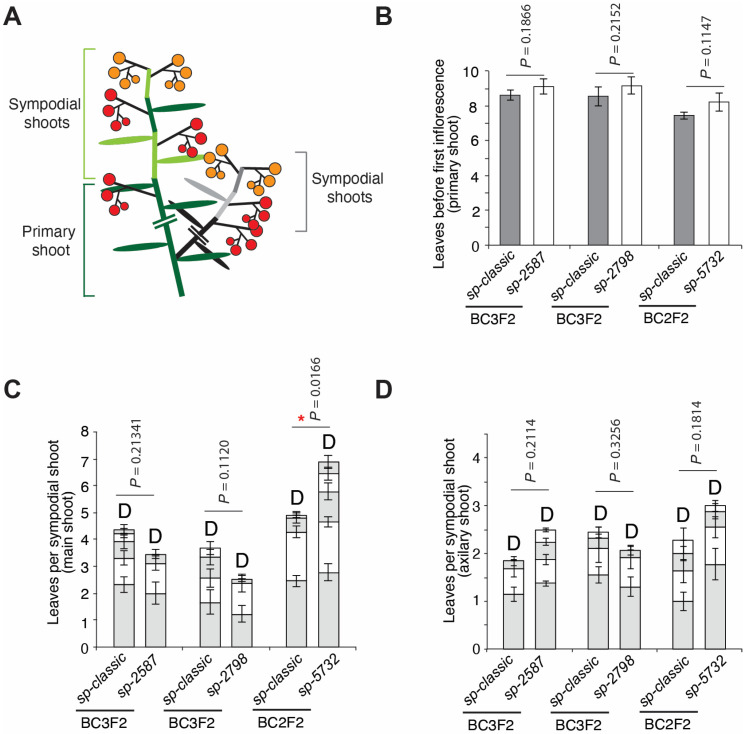
Comparison of flowering time with leaf numbers on the primary and sympodial shoot among *sp* alleles. (**A**) Diagrams depicting sympodial shoot growth on the main and axillary shoot in determinate tomato. Alternating dark green and light green bars represent the primary and successive sympodial shoots in main shoot. Alternating black and gray bars indicate the successive sympodial shoots in an axillary shoot. Ovals on the colored bars indicate leaves produced in each shoot. Red- and orange-colored circles represent maturing fruits. (**B**) Quantification and comparison of flowering time in the main shoot in three *sp* mutant alleles. *sp-classic* segregated from BC2 or BC3F2 generation was used as the control. (**C**,**D**). Quantification and comparison of leaves produced by five successive sympodial-shoot initially produced by primary shoot (**C**) and by axillary shoot (**D**). Mean values (± s.d) of leaves produced by each sympodial shoot (alternative grey and white color bar) were compared to those for *sp-classic* and *sp* allele segregated. The sum of the leaves from five sympodial shoots was used to test statistical significance in (**C**,**D**). D, terminated shoot; *p* values determined via two-tailed, two-sample t-test; * *p* value < 0.05.

**Figure 5 ijms-23-07149-f005:**
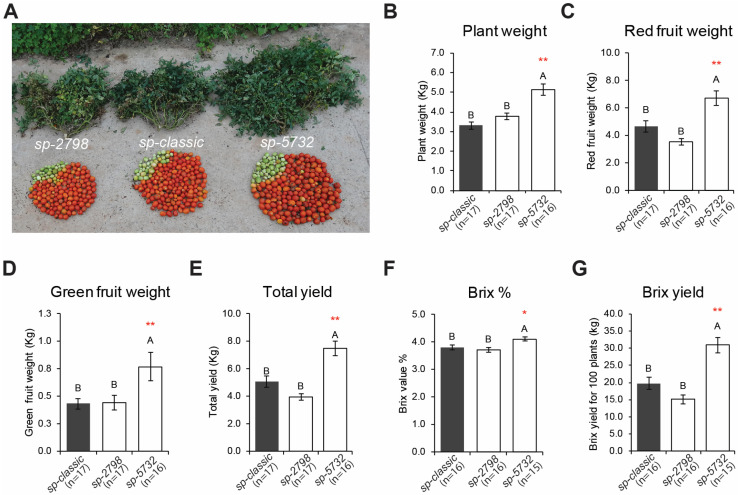
Variations in crop productivity induced by *sp* alleles at field. (**A**) Representative plant size and yield from *sp-classic* as control, *sp-2798*, and *sp-5732*. (**B**–**G**) Statistical comparisons of mean values (±S.E.M.) for plant weight (**B**), red fruit weight (**C**), green fruit weight (**D**), total yield (**E**), Brix (**F**), and Brix yield (**G**) from *sp-classic* (black bars), *sp-2798*, and *sp-5732* (white bars). Asterisks indicate significantly different yields. Different letters indicate significant differences between samples according to a one-way ANOVA followed by Tukey’s HSD post-hoc test (*p* < 0.05). Asterisks indicate significant differences with *sp-classic* by Tukey–Kramer test; * *p* value < 0.05, ** *p* value < 0.01. n, number of replicates.

**Figure 6 ijms-23-07149-f006:**
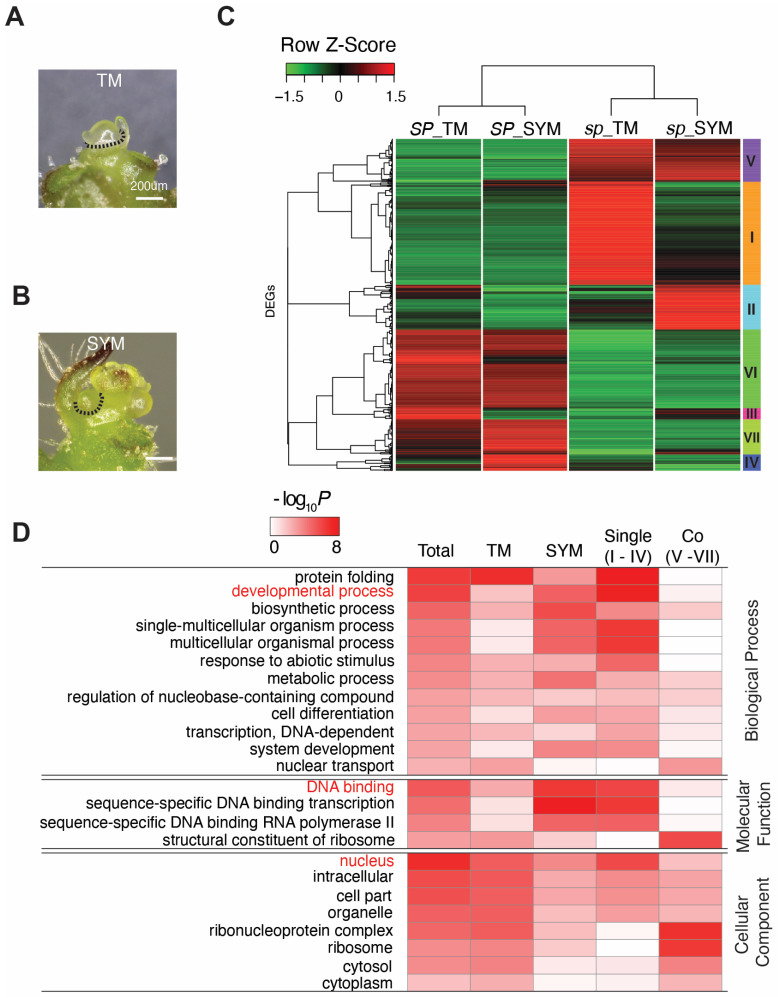
Hierarchical clustering analysis and GO term enrichment assay using DEGs between *SP* and *sp*-classic. (**A**,**B**) Microdissection of the TM stage (**A**) and SYM stage (**B**) was used for RNA extraction [24]. Dashed lines indicate dissected tissue lines. (**C**) Hierarchical clustering of DEGs at the TM and SYM stages between *SP* and *sp-classic*. Clustering is visualized by heatmap, and seven clusters (I–VII) are grouped based on the dendrogram. The FPKM expressions are normalized to row-wise Z-Scores. (**D**) Enrichment of gene ontology functional analysis of DEGs. -Scaled -log_10_^(*p* values)^ are shown in the heat map (Appendix A). ‘Single’ indicate DEGs showing differential expression in only one meristem stage. ‘Co’ indicate DEGs on both meristem stage. Red fonts indicate the major terms in functional analysis of ‘Single’ DEGs group.

**Figure 7 ijms-23-07149-f007:**
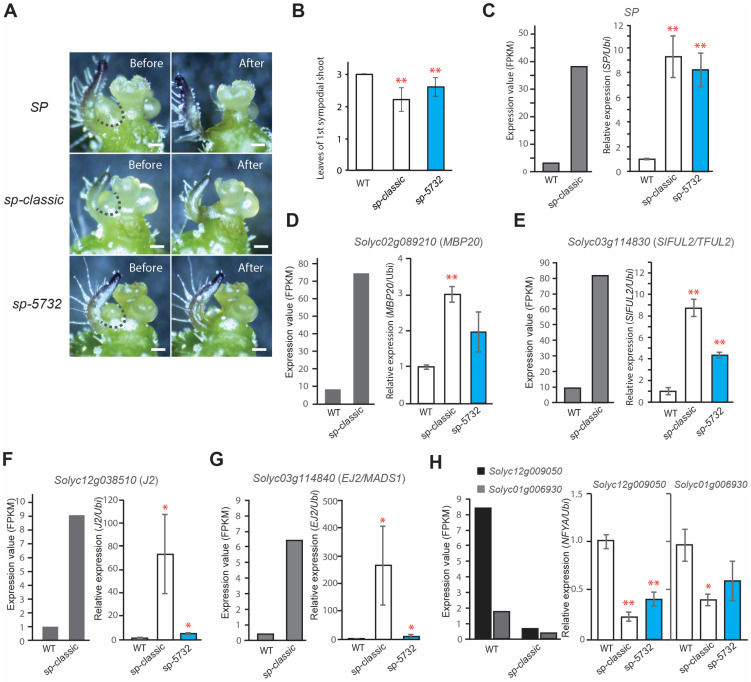
Quantification of molecular status of *sp-5732* SYM using biomarkers. (**A**) Microdissection of SYM from *SP* as the indeterminate control, *sp-classic* as the determinate control, and *sp-5732*. Dashed lines indicate dissected tissue lines on the intact SYM (Before). ‘After’ indicates the shoot apex image after SYM dissection. Scale bars, 100um. (**B**) Comparison of leaf production in the first SYM among three genotypes. (**C**–**H**) Comparisons of *SP* (**C**), *MBP20* (**D**), *SlFUL2* (**E**), *J2* (**F**), *EJ2* (**G**), and *Solyc12g009050* and *Soly01g006930* (**H**) expression in SYM detected by qRT-PCR. Gray bars indicate normalized read counts of each gene from tomato SYM RNA-seq data analysis. Blue bars indicate the values of leaf number and expression. Relative mRNA levels of each gene were normalized to the level of *Ubiquitin* mRNA. Statistical comparisons of the mean values (±s.d) were conducted with at least two biological replicates. *P* values were determined via two-tailed, two-sample *t*-test; * *p* < 0.05; ** *p* < 0.01.

## Data Availability

Not applicable.

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
