# Peer review of "Newly Discovered Alleles of the Tomato Antiflorigen Gene SELF PRUNING Provide a Range of Plant Compactness and Yield"

_ijms, 2022, doi:10.3390/ijms23137149_

Round 1
Reviewer 1 Report
The study by Kang et al describes a new variant of the tomato SP gene that is not as severe as the sp-classic allele in causing determinacy. Plants with this variant have extra sympodial units and, consequently, more biomass and greater yield. This allele was discovered by delving into the core collection of tomato cultivars (CC) and identifying determinate or semi-determinate lines that were not caused by the sp-classic allele. They also describe three sp variants from this collection, although the bulk of the study is focused on one variant, sp-5732. An expression profiling analysis was then carried out comparing the new sp allele with indeterminate SP plants
Overall the paper is well written (with the exception of some vague and cryptic statements, and a section on expression profiling described below), and the experimental approach is fairly straightforward. The upshot of this study is that they have identified a new sp allele that causes a change in plant architecture by the addition of extra sympodial units, as would be expected for a less severe sp allele. Of course, this has been described previously, but this does appear to be a new natural variant. And the finding that there is an increase in plant productivity is not surprising - of course there will be increased biomass if additional SYM units are added to the plant. The authors seem to imply a great significance to this result when in fact it is totally expected. In fact it is the basis of the initial selection process.
- The weakest part of this study is the section on the comparison of the molecular states of differentially expressed genes (section 2.3). The inclusion of the expression profiling study, comparing SP and sp-classic, is confusing and present some major problems. First, what is the connection with the sp-classic allele? It is not clear how this fits into the narrative of this paper. Second, the description of the differentially expressed genes seems to apply more significance to the findings than is warranted. The author describe them as new biomarkers, but it is not at all clear what this means and how this information can be applied to the selection of enhanced cultivars.
Third, I am not at all clear on how they selected the meristematic tissues for isolating RNA for this analysis. (By the way, what does TM mean? It is never described, but I assume it means terminal meristem?). Figure 6A and B shows the selected meristems used, but how do they know these meristems will produce TMs or SYMs? This needs to be described more clearly. For example. lines 249-252: what does this mean: “we identified 811 ‘TM DEGs’ 250 between TMs of the genotypes, and 520 ‘SYM DEGs’ between SYMs of the genotypes; and 251 a total of 984 ‘total DEGs’ between TM and SYM paired with genotypes…”. It seems that this part of the paper was written by someone that is analysing the data, but making no attempt to explain its significance or connect it to the main narrative of this study. Subsequent descriptions of biological functions via GO analysis offers up only generic terms with no contribution to illuminating what these data could mean.
Overall, the DEG section needs to be totally re-written for clarity and with an eye to link these findings to the main theme of this paper. As it stands it comes across as a free-wheeling add-on that simply pads the paper with more data. More importantly, what is the biological significance of this information?
- Another problem is that they assume the observed variation in determinacy is due to SP alleles only - however, they might have selected for modifiers of SP as well, which maybe could be a variant of another homolog - CETS is a large gene family. They even say they have no molecular explanation for sp-classic and their new allele (lines 352,353). Therefore they should acknowledge at the very least that there is a distinct possibility that they are observing the effects of modifier loci that just happen to be in the sp-5732 background.
- Finally, the last paragraph (lines 407-411) simply makes no sense. What approach are they referring to? How does this lead to marker assisted breeding? And the last sentence seems to be a random assortment of words with no real meaning.
Other points:
- line 16: is it correct to say that heterozygous sft mutants in the sp-classic background has a “heterosis-like effect”?
- lines 64-66: The SP protein does not form a FAC. Also, it is more correct to designate SP and SFT as members of the PEBP family - florigen family proteins are associated with floral induction.
- line 115-116: if you select for tomato lines from the CC it is not random.
- lines 115-165: This description of the discovery of new sp alleles is a bit of a “long walk”. The section is written in a confusing way, and it is difficult to parse the path they have taken to get to the final result; i.e., the molecular differences in these new alleles. This section needs to be reorganized for greater clarity. Related to this, line 194: they introgress 4 alleles into M82. What is the 4th allele? Maybe I missed it, but definitely re-writing will clarify what they are trying to do.
- line 129: confusing to say “other five lines”. There are 14 lines in total, so there are actually 9 lines left to consider.
- line 287: “expression”, not “expressional”.
- line 350: “distinct:, not “distinguished”.
- Materials and Methods, section 4.2: The word “ref” is in place of actual references.
- ref 14: has no journal or date information.
- Figure 4C and 4D are very confusing. What do the grey and white sub-bars within the bars mean? There is no explanation of this that I can see, in the text or the legend.
- Figure 5: This seems to be rather extraneous. As mentioned above, it is obvious that increasing SYM units will increase plant productivity. Why add all this information about fruit yield (red and green), total yield and Brix (!). This is an extreme example of over-padding the paper with non-informative data. And this could be included as supplemental information, if need be.
Figure 7A: to what does “before” and “after” refer? There is no description of this in the text or legend.
Author Response
Thank you for your time and efforts, as well as the valuable comments, it is appreciated. We have highlighted the changes within the manuscript.
And the finding that there is an increase in plant productivity is not surprising - of course there will be increased biomass if additional SYM units are added to the plant. The authors seem to imply a great significance to this result when in fact it is totally expected. In fact it is the basis of the initial selection process.
Response
We appreciate that you raised important questions about the significance of this result. As you know, the Core collection is a subset of a large germplasm collection that contains accessions chosen to represent the genetic variability of the germplasm collection. Molecular breeders have taken advantage of this to isolate new mutant alleles for gaining or improving traits of interest.
We suggest this manuscript is the first report that tomato yield increase was achieved using newly discovered sp alleles isolated from Tomato core collections. So far, no reports of suppressing sp (weak sp) are available, only hybrid studies (Nat. Genet. 2010, 42, 459–463; Nat. Genet. 2014, 46, 1337–1342; Cell 2017, 171, 470-480).
Crop yield improvement is a very important topic for plant scientists. New genetic variants showing high yield phenotypes have rarely been reported using classic breeding, including molecular breeding because variants that can maximize crop yields were not detected or may be absent from natural populations.
In this manuscript, the hypothesis of ‘suppression of sp’ in previous reports suggested the additional production of sympodial shoots in main and axillary stems resulting in a yield increase in sft/+ sp and sft/+ ssp/+ sp. Because nobody has evaluated whether the suppression of sp would translate to weak sp mutant alleles and an increase in tomato production, we selected the CC lines showing determinate growth. We isolated sp mutants from determinate CC lines and identified all the sp mutations, such as amino acid substitution, and gene expression to identify the new sp allele with a higher tomato yield and most optimal sympodial growth compared with the classical self-pruning mutant.
In addition to identifying sp alleles and tomato yield trials, our primary interest was not to classify and compare the DEGs between transcriptomes of SP and sp-classic but rather to use the biomarker isolated from DEGs as a quantitative molecular phenotyping tool to determine if there are changes in meristem maturation caused by weak sp allele before gross morphological differences in shoot architecture become apparent, resulting in high tomato production.
- The weakest part of this study is the section on the comparison of the molecular states of differentially expressed genes (section 2.3). The inclusion of the expression profiling study, comparing SP and sp-classic, is confusing and present some major problems.
First, what is the connection with the sp-classic allele? It is not clear how this fits into the narrative of this paper.
Response
We appreciate your comment and agree with the observation. Therefore, we added a few sentences and edited the text to explain why the expression profiling study comparing SP and sp-classic was performed. We have highlighted the changes within section 2.3..
Second, the description of the differentially expressed genes seems to apply more significance to the findings than is warranted. The author describe them as new biomarkers, but it is not at all clear what this means and how this information can be applied to the selection of enhanced cultivars.
Response
Thank you for your comment, which is reasonable. However, our primary interest was not only to classify and compare the DEGs but to use biomarkers isolated from DEGs as quantitative molecular phenotyping tools to determine if there are changes in meristem maturation caused by weak sp allele before gross morphological differences in shoot architecture become apparent. In detail, we isolated six biomarkers from transcriptome analysis which were used to quantify the molecular states in the SYM of sp-5732 between indeterminate (SP) and determinate sympodial growth (sp-classic). The six markers were associated with two genes of the potential targets of FAC, two genes physically interacting with MADS TFs controlled by FAC, and two functionally unknown and downregulated TFs. Especially, the last two unknown TFs showed expression values between the expression of SP and sp-classic, which are similar to results with the other four biomarkers, confirming the difference in the molecular state of sp-5732's SYM.
Third, I am not at all clear on how they selected the meristematic tissues for isolating RNA for this analysis. (By the way, what does TM mean? It is never described, but I assume it means terminal meristem?).
Figure 6A and B shows the selected meristems used, but how do they know these meristems will produce TMs or SYMs? This needs to be described more clearly. For example. lines 249-252: what does this mean: “we identified 811 ‘TM DEGs’ 250 between TMs of the genotypes, and 520 ‘SYM DEGs’ between SYMs of the genotypes; and 251 a total of 984 ‘total DEGs’ between TM and SYM paired with genotypes…”.
Response
We appreciate your comments and agree. Therefore, we added the details about TM and SYM defined in previous studies. We also explained how we captured the TMs and SYMs from two SP alleles for profiling in the manuscript by revising section 2.3..
It seems that this part of the paper was written by someone that is analysing the data, but making no attempt to explain its significance or connect it to the main narrative of this study. Subsequent descriptions of biological functions via GO analysis offers up only generic terms with no contribution to illuminating what these data could mean.
Overall, the DEG section needs to be totally re-written for clarity and with an eye to link these findings to the main theme of this paper. As it stands it comes across as a free-wheeling add-on that simply pads the paper with more data. More importantly, what is the biological significance of this information?
Response
We appreciate your comments and agree. Therefore, we rewrote the part of DEG, clustering and GO analysis and described the biological significance of this information in the revised manuscript. We have highlighted the changes within section 2.3.
- Another problem is that they assume the observed variation in determinacy is due to SP alleles only - however, they might have selected for modifiers of SP as well, which maybe could be a variant of another homolog - CETS is a large gene family. They even say they have no molecular explanation for sp-classic and their new allele (lines 352,353). Therefore they should acknowledge at the very least that there is a distinct possibility that they are observing the effects of modifier loci that just happen to be in the sp-5732 background.
Response
Thank you for your comment. We understand there might be modifiers of SP in the mutant alleles. To remove the modifier issues, we did four backcrossings with sp-classic and genotyped the sp alleles with specific genotyping markers. We examined sympodial growth from sp-5732 and sp-classic segregated in the BC2F2 and BC3F2 generations. Data from both were identical, meaning that the sympodial growth cannot be genetically affected by a modifier of SP (see Figures 4 and 7B and C).
Because we didn’t perform any protein function analysis with the sp mutations, we mentioned it in the discussion: “Although we could not clearly explain the differences in functional defects among sp alleles, including between sp-5732 and sp-classic, ….” Exploring why the mutation causes a weak function should be a future study.
- Finally, the last paragraph (lines 407-411) simply makes no sense. What approach are they referring to? How does this lead to marker assisted breeding? And the last sentence seems to be a random assortment of words with no real meaning.
Response
Thank you for your comment. We edited the last paragraph. Three sp alleles have been used for breeding modern tomato cultivars by phenotyping; however, studies on sp allele variation are lacking. Based on the new sp alleles, breeders can select the most optimal plant determinacy by genotyping sp alleles. The changes are highlighted in lines 443–448.
- line 16: is it correct to say that heterozygous sft mutants in the sp-classic background has a “heterosis-like effect”?
Response
Thank you for the comment. We edited the sentence.
- lines 64-66: The SP protein does not form a FAC. Also, it is more correct to designate SP and SFT as members of the PEBP family - florigen family proteins are associated with floral induction.
Response
Thank you for the comments. We partially agree with your comment. We suggest that SP protein could be a cofactor for transcription because SP interacts with 14-3-3 (Plant Cell 2001, 13, 2687–2702) and 14-3-3 interacts with SSP (Nat. Genet. 2014, 46, 1337–1342), which means potentially SP forms SP:14-3-3:SSP similar to FAC. Based on this concept, SP protein molecularly controls gene expression related to floral phase-changing developmental genes. As described in the results and discussion section, the two genes that are the potential downstream genes of FAC that function in floral phase transition were highly downregulated in the sp classic.
- line 115-116: if you select for tomato lines from the CC it is not random.
- lines 115-165: This description of the discovery of new sp alleles is a bit of a “long walk”. The section is written in a confusing way, and it is difficult to parse the path they have taken to get to the final result; i.e., the molecular differences in these new alleles. This section needs to be reorganized for greater clarity. Related to this, line 194: they introgress 4 alleles into M82. What is the 4th allele? Maybe I missed it, but definitely re-writing will clarify what they are trying to do.
Response
Thank you for your great comment. As you suggested, we partially reorganized the paragraphs about isolating new sp alleles from CC lines, using the step-by-step process description to identify sp mutant from CC lines. The description order is how we selected the CC lines showing determinate growth and follows how we isolated sp mutants from determinate CC lines and then how we identified all the sp mutations, such as amino acid substitution, and expression.
Thank you for your comment on the number of sp alleles. We fixed the sp allele number as three alleles.
- line 129: confusing to say “other five lines”. There are 14 lines in total, so there are actually 9 lines left to consider.
Response
Thank you for the comments. Edited.
- line 287: “expression”, not “expressional”.
Edited.
- line 350: “distinct: not “distinguished”.
Edited.
- Materials and Methods, section 4.2: The word “ref” is in place of actual references.
- ref 14: has no journal or date information.
Response
Thank you for the comments. Edited.
- Figure 4C and 4D are very confusing. What do the grey and white sub-bars within the bars mean? There is no explanation of this that I can see, in the text or the legend.
Response
Thank you for the comments. Text was added to Figures 4C and D legends.
- Figure 5: This seems to be rather extraneous. As mentioned above, it is obvious that increasing SYM units will increase plant productivity. Why add all this information about fruit yield (red and green), total yield and Brix (!). This is an extreme example of over-padding the paper with non-informative data. And this could be included as supplemental information, if need be.
Response
Thank you for the comments. However, we could not agree with these comments. It is true that the number of sympodial shoots is one of the traits related to tomato yield, but this should be carefull to not to be overemphasized. Nobody can say that more sympodial shoots will increase crop productivity without yield trials showing consistently positive increases Figure 5 is a basic list to prove tomato yield improvement which is the most crucial data in this paper reflecting sp-5732 optimization of tomato productivity using SP alleles..
Figure 7A: to what does “before” and “after” refer? There is no description of this in the text or legend.
Response
Thank you for the comments. The legend for Figure 7A were added to explain the “before” and “after”.

Reviewer 2 Report
Review ijms-1709147
The authors reported alleles of the tomato antiflorigen gene SELF PRUNING provided some different attributes of the fruit. The results provided some useful information. However, some of the expression and discussion should be improved to enhance the quality.
Figures 1 and 4a: not just schematic images but also the real photos of the samples can be presented.
Figure 5: E-G the description of these parameters should be enhanced in the corresponding methodology section.
Section 2.3: more rationale should be provided on selecting several parameters. For instance, log2-fold change and FDR. Whether this selection matches with previous reports on plant-based food materials? For example, Food Chemistry, 286,87-97.
The discussion should be enhanced, especially on the relationship between attributes and metabolomics of the plant/tomato. For instance, Metabolomics, 11, 302-311.
Author Response
Thank you for your time and efforts, as well as the valuable comments, it is appreciated. We have highlighted the changes within the manuscript.
The authors reported alleles of the tomato antiflorigen gene SELF PRUNING provided some different attributes of the fruit. The results provided some useful information. However, some of the expression and discussion should be improved to enhance the quality.
Response
Thank you for your time, efforts, and valuable comments; it is appreciated. We have highlighted the changes within the manuscript.
We partially reorganized the paragraphs about isolating new sp alleles from CC lines, using the step-by-step process description to identify sp mutants from CC lines. The description order is how we selected the CC lines showing determinate growth and followed how we isolated sp mutants from determinate CC lines and identified all the sp mutations, such as amino acid substitutions, and determined expression. We added a few sentences to explain why the expression profiling study comparing SP and sp-classic was performed in the revised manuscript.
Figures 1 and 4a: not just schematic images but also the real photos of the samples can be presented.
Response
Thank you for the comments. We added the photos to Figure 1, but not Figure 4a due to the time limitation of generating mature plants.
Figure 5: E-G the description of these parameters should be enhanced in the corresponding methodology section.
Response
Thank you for the comments. In the Materials and Methods section, we added a paragraph describing the methods for measuring red and green fruits, total yield, Brix, and Brix.
Section 2.3: more rationale should be provided on selecting several parameters. For instance, log2-fold change and FDR. Whether this selection matches with previous reports on plant-based food materials? For example, Food Chemistry, 286,87-97.
Response
Thank you for the comments. We used the parameters that have been used for DEG analysis in previous reports, and a reference was added.
The discussion should be enhanced, especially on the relationship between attributes and metabolomics of the plant/tomato. For instance, Metabolomics, 11, 302-311.
Response
Thank you for the comments. We agree that the metabolic process term is one of the highly enriched terms from GO analysis. However, we think it would be better to focus on the genes related to the developmental process and transcriptional regulation terms. The reason is that our primary interest in DEGs in this manuscript was to use the biomarker isolated as a quantitative molecular phenotyping tool to determine if there are changes in meristem maturation caused by weak sp allele before gross morphological differences in shoot architecture become apparent.

Reviewer 3 Report
Minor revision:
1. Figure 4B: Differential significance should be added.
2. Figure 4C,D, Line 325, 326: 'P' should be in italic.
Author Response
Thank you for your time and efforts, as well as the valuable comments, it is appreciated. We have highlighted the changes within the manuscript.
Figure 4B: Differential significance should be added.
Response
Thank you for the comments. We added the p value to each comparison in Figure 4B.
Figure 4C,D, Line 325, 326: 'P' should be in italic.
Response
Thank you for the comment. Edited.